# A Roadmap for Potential Improvement of Newborn Screening for Inherited Metabolic Diseases Following Recent Developments and Successful Applications of Bivariate Normal Limits for Pre-Symptomatic Detection of MPS I, Pompe Disease, and Krabbe Disease

**DOI:** 10.3390/ijns8040061

**Published:** 2022-11-15

**Authors:** Kabir Jalal, Randy L. Carter, Amy Barczykowski, Shunji Tomatsu, Thomas J. Langan

**Affiliations:** 1Department of Biostatistics, Population Health Observatory, School of Public Health and Health Professions, University at Buffalo, Buffalo, NY 14214, USA; 2Nemours/Alfred I. DuPont Hospital for Children, Wilmington, DE 19803, USA; 3Department of Biological Sciences, University of Delaware, Newark, DE 19797, USA; 4Department of Pediatrics, Thomas Jefferson University, Philadelphia, PA 19107, USA; 5Department of Pediatrics, Graduate School of Medicine, Gifu University, Gifu 501-1194, Japan; 6Department of Pediatrics, Shimane University, Matsue 690-0823, Japan; 7Department of Neurology, School of Medicine and Biomedical Sciences, University at Buffalo, Buffalo, NY 14214, USA

**Keywords:** MPS1, detection, Krabbe, Pompe, bivariate normal limits, glycosaminoglycans, alpha L-iduronidase, galactocerebrosidase, psychosine, α-glucosidase, creatine

## Abstract

The mucopolysaccharidoses (MPS), Pompe Disease (PD), and Krabbe disease (KD) are inherited conditions known as lysosomal storage disorders (LSDs) The resulting enzyme deficiencies give rise to progressive symptoms. The United States Department of Health and Human Services’ Recommended Uniform Screening Panel (RUSP) suggests LSDs for inclusion in state universal newborn screening (NBS) programs and has identified screening deficiencies in MPS I, KD, and PD NBS programs. MPS I NBS programs utilize newborn dried blood spots and assay alpha L-iduronidase (IDUA) enzyme to screen for potential cases. Glycosaminoglycans (GAGs) offer potential as a confirmatory test. KD NBS programs utilize galactocerebrosidase (GaLC) as an initial test, with psychosine (PSY) activity increasingly used as a confirmatory test for predicting onset of Krabbe disease, though with an excessive false positive rate. PD is marked by a deficiency in acid α-glucosidase (GAA), causing increased glycogen, creatine (CRE), and other biomarkers. Bivariate normal limit (BVNL) methods have been applied to GaLC and PSY activity to produce a NBS tool for KD, and more recently, to IDUA and GAG activity to develop a NBS tool for MPS I. A BVNL tool based on GAA and CRE is in development for infantile PD diagnosis. Early infantile KD, MPS I, and PD cases were pre-symptomatically identified by BVNL-based NBS tools. This article reviews these developments, discusses how they address screening deficiencies identified by the RUSP and may improve NBS more generally.

## 1. Introduction

The mucopolysaccharidoses (MPS) are a group of lysosomal storage disorders (LSDs), a class of over forty conditions, which are characterized by excessive accumulation of complex lipids, glycoproteins, glycosaminoglycans (GAGs), and other macromolecules within the lysosome [1,2]. The results of metabolic disturbances caused by enzymatic deficiencies in LSDs vary symptomatically but often present multi-systemic clinical symptoms, including neurological, organ, and skeletal challenges [2]. Most LSDs comprise multiple phenotypic variants. The most severe phenotypes occur in the first months of life. For MPS, the unattenuated Hurler’s phenotype (MPS I) has the onset in the first year of life. In general, LSD phenotypes with earlier ages at onset are more severe, and these conditions can cause permanent dysfunction in affected systems [1,2].

Pompe disease (PD) and Krabbe disease (KD) are neurodegenerative disorders that more broadly are characterized by specific enzymatic deficiencies. PD causes glycogen storage in skeletal and cardiac muscle [3,4]. KD primarily affects the white matter in the brain and peripheral nerves, with enzyme deficiency disrupting myelin production, degrading the protective myelin sheath around nerve endings and causing progressive symptoms [5,6,7]. Like LSDs generally, PD and KD phenotypes with earlier ages at onset of symptoms typically experience a more severe disease course, and infant mortality rates are high.

MPS, PD, and KD are generally progressive, and treatment has historically been limited to symptom management, with enzyme replacement therapy (ERT) being employed for only MPS and PD clinically in 1991 [4,8]. Hematopoietic stem cell transplantation (HSCT) offers the greatest improvement in clinical outcomes and attenuation of disease course for these three disorders, with several thousand transplants since the 1980s, but is reliant on the identification of diseased patients pre-symptomatically to maximize the efficacy of the treatment [9].

Considering the risks of infection and other complications associated with HSCT, and the difficulty with the reliable prediction of disease phenotypes, it is clear that improved pre-symptomatic identification of LSD patients is one of the most critical challenges facing newborn screening (NBS) programs and the LSD research community [10]. While there is no United States federal body enforcing standardized NBS, the Advisory Committee on Heritable Disorders in Newborns and Children (ACHDNC) maintains a collection of diseases collectively known as the Recommended Uniform Newborn Screening Panel (RUSP) [11,12]. For many diseases, the anxiety placed upon families with children undergoing screening procedures is a significant risk associated with NBS programs [10]. This is the crux of the “false positive problem” that would benefit from improved pre-symptomatic identification of diseased patients.

However, long term parental anxiety may not be a significant problem in all cases in which false positive results are given [13]. For example, anxiety seems to become minimal after six months after false positive results for cystic fibrosis [14]. It has also been suggested recently that improved second tier tests, third tier molecular tests, and analytical tools of the sort that we propose here can diminish parental anxiety [15].

Multivariate normal distribution theory allows for estimation of (1 − α)100% prediction regions [16]. Thus, we have developed (1 − fp)100% *Bivariate Normal Limits* (BVNL), where fp is a pre-specified tolerable false positive rate, as a post-analytical tool to test for early childhood KD [5]. The resulting NBS test was refined based on improved input data to produce a finalized NBS tool, based on GaLC enzyme activity and psychosine (PSY) concentration measures from newborn dried blood spots (DBS). The tool was retrospectively evaluated for pre-symptomatic identification of patients with early childhood KD who would most benefit from HSCT [6].

The BVNL methodology was subsequently applied to MPS I with apparent success, with the resulting BVNL tests yet to be validated [7]. Applications of the BVNL tool can be performed on biomarkers already being collected during many newborn screening protocols, minimizing additional cost or effort and without need for proprietary software or submission of data to a third party. This review article summarizes BVNL methodology and its applications to NBS for KD, PD, and MPS I to provide a roadmap for the development and application of BVNL-based NBS tests for inherited metabolic diseases more generally.

## 2. History and Current Practices in NBS for MPS, PD and KD

The advent of tandem mass spectrometry (MS/MS) in the 1990s enabled laboratories to conduct more efficient newborn screening [17]. Coupled with the establishment of the RUSP, this has enabled several pilot NBS programs for MPS I, the first of which began in 2008 in Taiwan [18,19,20,21,22,23,24,25,26,27]. GAG measurements, IDUA enzyme assays, and molecular diagnosis have all been implemented for MPS I screening [28]. Currently, universal NBS programs that screen for MPS I are conducted in several states, including Illinois, Washington, Kentucky, and New York [21,23,26,29]. Globally, NBS programs screen for MPS I in countries including Italy, Taiwan, Japan, and Brazil [22,24,30,31]. Modern NBS for PD began in 2005 in Taiwan and utilized fluorescence assays [32,33,34,35]. Further NBS programs in Japan, Italy, Hungary, Germany, Colombia, Austria relied on tandem MS/MS as well as fluorescence assays [32,33,36,37,38,39,40,41]. In the United States, programs in Washington, New York, Illinois, Kentucky, Mississippi, and Pennsylvania utilized tandem MS/MS, with Missouri NBS programs implementing digital microfluidics screening method [23,32,33,42,43,44]. NBS efforts for KD have been ongoing in several US states [45,46,47].

The efficacy of these programs in the previous decade have been mixed at best. For MPS I, NBS programs in the United States, Italy, and Taiwan have seen PPV ranging from 0–50% [21,22,23,24,26]. For PD, NBS programs abroad in Taiwan and Austria achieved PPV of 63.4% and 80% but subsequent Italian and Hungarian studies could only achieve 6%, while US efforts in Illinois, Missouri, Washington, and New York achieved PPV ranging from 3–21% [33,34,37,38,41]. KD programs likewise have been insufficient; NBS programs in New York State could only achieve a PPV of 1.4% prompting reservations of proliferated NBS, while Illinois screening programs using PSY and post-analytical software (Collaborative Laboratory Interpretive Reports, https://clir.mayo.edu (accessed on 4 October 2022) could only achieve 40% PPV without genetic testing [29,45,46,47].

Latest efforts for MPS I and KD, however, have begun to show increasing promise. MPS I screening has shown improved PPV using a two-tier testing approach incorporating DS and HS, with PPV rates of 74% in the US and 100% in retrospective studies [48,49]. While PPV rates in current KD studies remain lower, recent studies have highlighted the improvement that genetic testing can offer, with Illinois achieving 100% PPV when genetic testing is used [47]. Indeed, recent consensus recommendations suggest genetic testing may reduce follow-up testing by 88% [50]. Regarding PD, however, a 2020 review highlights the difficulty of discriminating IOPD from pseudodeficiency [51].

US NBS programs now screen for at least 30 core disorders and up to 26 secondary disorders [52]. In the cases of MPS I and PD, these conditions have been added to the RUSP despite the significant screening deficiencies. MPS I/II are the only MPS disorders included on the RUSP, having their nomination confirmed in 2016 and 2022 [53,54]. PD was recommended for addition to the RUSP in 2013 and added as a core disorder in 2015 [25]. The AHCDNC completed a review of EIKD in 2010 and declined its inclusion on the RUSP, noting that while EIKD would benefit from early diagnosis and intervention, the possibility of substantial harm from screening and/or treatment precluded addition to the panel [55]. Indeed, all three conditions discussed here would benefit from improved pre-symptomatic identification of diseased newborns.

## 3. General BVNL Methods with Example Application to KD, MPS I, and PD

### 3.1. General Introduction to BVNL

BVNL methodology is based on well-established multivariate normal distribution theory [56]. Sir Francis Galton recognized that plots of the heights of parents and their adult children formed ellipses [57]. His depiction drove the development of a large portion of modern statistical techniques, ranging from regression and correlation to multivariate normal distribution theory and its two-dimensional special case, bivariate normal distribution theory [56,58].

In the context of disease diagnosis, these methods utilize a set of biomarker observations from disease-free patients to form a normal tolerance region that can be estimated by a prediction region. With sufficient normal observations, this prediction region contains a pre-specified portion, (1−α)100%, of the disease-free or normal population, where (1−α) represents the confidence that a randomly sampled individual from the normal population will fall in the region. The approximation improves with increasing sample size, as prediction regions converge to corresponding tolerance regions as sample size increases. While these techniques generalize to any number of variables, this discussion will center on the two-variable case where (1−α)100% prediction regions are ellipses.

For a diagnostic screen that utilizes a single predictive biomarker, this tolerance region would be a univariate (1−α)100% interval, and observations falling above or below a certain critical univariate threshold value for the biomarker would be considered a positive screen. Given a second predictive biomarker, using a second critical univariate threshold would improve diagnostic accuracy. However, given two biomarkers with a bivariate normal distribution that are informative about disease risk, simply using univariate intervals or thresholds does not account for the correlation between the two biomarkers and is, therefore, less efficient than a bivariate approach, resulting in additional false positives.

Reference ranges for single biomarkers measured from blood spots or urine analysis have long been used in NBS program protocols [10]. More recently, there has been increasing interest in two-tier testing, analyzing a second biomarker on the same NBS blood spot material used in an initial positive screen [59]. While separate univariate reference ranges for each can be useful in two-tier testing, statistical improvements in accuracy are achievable by using bivariate normal limits based on tolerance regions, as suggested above, provided the distribution of the biomarkers is at least approximately bivariate normal [5]. Furthermore, screening tools utilizing three or more biomarkers in concert may be improved by implementing multivariate normal limits, as these methods generalize to any number of biomarkers.

### 3.2. General Definition and Discussion of BVNL Screening Tests for Inherited Metabolic Diseases

The root causes of inherited metabolic diseases such as KD, PD, and MPS disorders are pathogenic genetic variants and related enzyme deficiencies/inactivity that result in an accumulation of toxic substances in cells and ultimately damage the central nervous system, organs, or tissue. The etiology of such diseases makes them particularly well-suited for NBS tests that are based on multiple biomarkers: a measure enzyme level or activity (or a monotone transformation of such a variable) and a measure of the toxic substance levels in the cells (or a monotone transformation such a variable). Transformations are selected, if necessary, so that the biomarkers are normally distributed. If the assumption of normality is met, a (1 − α)100% prediction ellipse can be estimated. Given biomarkers X and Y, and thresholds τ_1_ (indicating “low” values of X) and τ_2_ (indicating “high” values of Y), a BVNL NBS test employs the following prediction rule:
(1)Predict that the infant will experience clinical symptoms of the disease in early childhood if
the observed value of X is less than τ_1_,the observed value of Y is greater than τ_2_, andthe observed value of the pair (X, Y) falls outside the estimated (1 − α)100% prediction ellipse; and(2)Predict that the infant will not experience clinical symptoms during early childhood if any of the conditions a, b, or c above do not hold.

To specifically define a test for use, a NBS program must specify the values of τ_1_, τ_2_, and α.

α should be set at the largest false positive rate that is tolerable to the program, as α is the maximum possible false positive rate of a prediction rule that is defined as above. The actual false positive rate (FPR) will be less than α; how much less depends on the choice of τ_1_ and τ_2_. The smaller τ_1_ and larger τ_2_ are, the lower FPR will be. Given the choices of τ_1_, τ_2_, and α and a large sample of normal infants; a close estimate of the actual FPR can be computed theoretically or estimated by a simulation study. This is not necessary given the assumption of normality is met, as it is guaranteed that FPR < α; and one needs to simply set α at their maximum tolerable value of FPR to achieve acceptable control of the actual FPR. The normality assumption can be verified by inspection of the sample quantile-quantile plot of robust Mahalanobis squared distances versus the known quantiles of the Chi-square distribution with two degrees of freedom [56]. Then, given large n and consistent maximum likelihood estimates the resulting prediction ellipses will contain approximately (1 − fpr)100% of future normal observations [60].

Evaluating a diagnostic test for any disease requires estimation of six epidemiological parameters; Sensitivity (Sens), Specificity (Spec), Positive Predictive Value (PPV), Negative Predictive Value (NPV), False Positive Rate (FPR), and False Negative Rate (FNR); which are estimated as follows when a random sample of size N = *TP* + *FP* + *FN* + *TN* is available from the general population:(1)Sensitivity= TP(TP+FN)
(2)Specificity=TN(TN+FP)
(3)Positive Predictive Value=TP(TP+FP)
(4)Negative Predictive Value=TN(TN+FN)
(5)False Positive Rate=FP(FP+TN)=1−Specificity
(6)False Negative Rate=FN(FN+TP)=1 –Sensitvity
where *TP* represents the number of correct positive test results, *FP* the number of incorrect positive test results, *FN* the number of incorrect negative test results, and *TN* the number of correct negative test results.

In the case of rare diseases, small samples may leave one or more of these quantities inestimable. For some diseases, the necessary additional information is available in the form of disease prevalence estimates from the literature. However, if accurate disease prevalence estimates are available, PPV can be estimated using Bayes’ Rule as follows.
(7)PPV=Sens(Sens+FPR*O)
where O=(1−Prev)/Prev, and Prev denotes the prevalence of the disease in the general population. Thus, Equation (7) is the odds that a randomly sampled infant from the general population has the disease. So, if the prevalence of the disease is known or externally estimated, then a valid estimate of PPV can be calculated by substituting the estimate of sensitivity in Equation (1) and the calculated value of O into Equation (7), provided that *FP* ≠ 0. Analogous statements hold for NPV but are not discussed here, as Equation (7) is particularly relevant in BVNL applications to KD and MPS I presented below.

A beauty of BVNL NBS tests is that FPR is knowable or even can be fixed at an acceptable level by one’s choice of τ_1_, τ_2_, and α. Thus, PPV can be properly estimated even when FP = 0, if an estimate of prevalence is available. Furthermore, even without performing the difficult mathematical tasks of theoretically calculating FPR, given chosen values of τ_1_, τ_2_, and α, or of choosing values of τ_1_, τ_2_, and α that ensure an acceptably low pre-specified; we have the following lower bound on PPV:(8)PPV>Sens/(Sens+α*O),
because it is mathematically guaranteed that the FPR of a BVNL test is less than α.

### 3.3. Review of an Application of a BVNL NBS Test for KD

The need for a bivariate approach to EIKD NBS has been established by examination of univariate normal limits of GaLC enzyme, concluding that while depleted GaLC enzyme levels were indicative of EIKD, they could not solely determine phenotype [61]. After interest in PSY re-emerged, measurements of GaLC and PSY were used successfully in an initial BVNL approach, although the lack of simultaneous GaLC/PSY measurements from a normal population limited studies to investigations of the potential benefits of a bivariate approach [59,62] and ad hoc development and application of the first BVNL NBS test for KD [5].

The results were positive, and work began to collect simultaneous GaLC/PSY measurements from healthy newborns, which was necessary for fully rigorous development. In October 2016, data from 166 NBS dried blood spots, as well as 15 affected KD cases with symptom onset prior to 29 months, were utilized to further develop a BVNL test for KD screening [6]. This involved standardizing and centering natural-log transformations of GaLC and PSY determinations on deriving a (1–10^−6^)100% prediction ellipse for z-scores that is portable to any NBS program. The values of τ_1_ = −2.90, τ_2_ = 2.90, and α = 10^−6^ were chosen. These settings corresponded roughly to a FPR of 10^−7^. Figure 1 below shows the resulting ellipse and results of the application of the resulting BVNL test to the normative and diseased samples.

For this FPR only one falsely predicted early childhood case of KD is expected to occur in every 10 million newborns [6]. The rough approximation that FPR = 10^−7^ for the above settings of τ_1_, τ_2_, and α was confirmed by Monte Carlo simulation results. By generating 100,000,000 observations from the estimated distribution of normal newborns illustrated by the ellipses in the figure above and tabulating the number of observations falling in the abnormal region, a simulated FPR was estimated and report to be 1.1 per 10 million newborns, very close to the roughly approximated 1 per 10 million [5]. This FPR corresponds to one expected false positive every 2.5 years if every US newborn were screened [6].

Langan et al. [6] reported an estimated sensitivity of 1.0 and specificity also of 1.0. An estimated PPV was obtained by substituting these estimates along with FPR =10^−7^, and *O* = 149,999 into Equation (8) above to obtain an estimated PPV of 98.5%, which far exceeds estimated PPV of previously employed test protocols that do not use BVNL [46]. *O* was calculated from a reported prevalence of 1 in 150,000 from the literature [63]. Efforts are currently underway for a prospective evaluation/validation of this BVNL test for KD.

Carter et al. showed that the BVNL test for KD performed better than any univariate test based on GaLC alone (Predict KD if X< τ_1_), a univariate test based on PSY alone (Predict KD if Y > τ_2_), and a bivariate test that predicts KD if X < τ_1_ and Y> τ_2_ (i.e., a bivariate test that is based on conditions a and b above, but not c) [64]. The operating characteristics of the BVNL test were better than those of the latter of these three tests because the BVNL test incorporates information about the shape of the distribution of (X, Y) points in the normal population to better identify abnormal observations. The BVNL offers a theoretically ensured improvement in FPR that did not increase FNR in the application.

### 3.4. Review of an Application of a BVNL NBS Test for MPS I

Similarly to deficient GaLC enzyme activity causing a harmful excess of PSY levels in KD patients, deficiency in IDUA enzyme leads to the harmful accumulation of GAGs. Kubaski et al. examined dried blood spots from 2862 NBS patients and 14 MPS cases, 7 of which were MPS I patients, demonstrating that certain GAGs may be beneficial in NBS programs as a first- or second-tier test [27]. Langan et al. considered IDUA enzyme and the GAG heparan ΔDi-NS [2-deoxy-2-sulfamino 4-O-(4-deoxy-α-L-*threo*-hex-4-enopyranosyluronic acid)-D-glucose] (HS) as part of BVNL approach to MPS I NBS [7].

Using 5000 normal newborns from Japanese screening efforts in the Gifu prefecture, BVNL prediction ellipses were calculated for univariately centered and standardized natural log values of HS and IDUA activity. This resulted in a (1–10^−7^)100% BVNL prediction ellipse. The values of τ_1_ = −3.62, τ_2_ = 1.90, and α = 10^−7^ were chosen. Thus, NBS observations with transformed IDUA less than −3.62 and transformed HS greater than 1.9 that fall outside the BVNL ellipse are test-positive BVNL. These thresholds and α–level result in roughly a FPR of 10^−8^. The MPS I BVNL plot is shown below in Figure 2, with seven MPS I cases and 12 pseudo-deficient normal newborns [7].

Langan et al. followed a similar simulation strategy as in their KD application to estimate specificity and PPV [6,7]. Ultimately, they report that this BVNL tool for MPS I yields one false positive in 100 million newborns tested, with a sensitivity of 100%, specificity of 99.999999%, and a PPV of 99.9%. Langan et al. conclude that the BVNL tool outperformed univariate threshold tests using IDUA and HS, and also the joint univariate test of both IDUA/HS [7].

### 3.5. Review of an Application of a BVNL NBS Test for Infantile PD

Following efforts with KD and MPS I, Langan et al. applied BVNL methods to biomarkers relevant to PD. In the case of PD, deficient GAA enzyme combined with creatine levels seemingly offer potential in the diagnosis of IOPD. While further refinement of the ellipse and additional testing is required on some referred patients before the ellipse itelf may be published, the resultant preliminary findings are discussed below.

Dried blood spots of 312,105 normal newborns from New York State screening programs were tested for CRE and GAA activity. This resulted in a (1–10^−8^)100% BVNL prediction ellipse. The values of τ_1_ = −4, τ_2_ = −1, and α = 10^−8^ were chosen. Thus, NBS observations with transformed GAA less than −4 and transformed CRE greater than −1 that fall outside the BVNL ellipse are test-positive BVNL. These thresholds and α–level result in roughly a FPR of 10^−9^. The PD BVNL test accurately identified seven known IOPD cases and 312,105 normal newborns. Four of the presumptively normal newborns were also identified as PD cases by the preliminary BVNL tool.

Unlike KD and MPS I, the presence of false positives on the preliminary PD ellipse allows for direct estimation of the achieved false positive rate and PPV using Equations (3) and (5). With four apparent false positives, the BVNL method achieves a FPR of 0.000013%, with 95% confidence interval of (0.0000003, 0.00003), and a PPV of 63.64%, with 95% confidence interval (35.21, 92.06). It should be noted that adjusting τ_1_ (the GAA threshold) to −7.5 would eliminate all false positives, although the risk of missing true cases increases. Although not as well performing as the BVNL for KD and MPS I, further refinements such as an alternative to CRE as a second-tier biomarker or different choice of transformation for GAA and/or CRE are currently being investigated. Nevertheless, these achieved values are improvements on several reported studies [32,33,36,37,38,39,40]. Further, compared to the BVNL implementation, using univariate thresholds alone as screening criteria would have resulted in a minimum of 299 additional false positives.

## 4. Discussion

The current article argues that BVNL NBS tests are appropriate for inherited metabolic diseases in general and presents two examples of very successful applications of them in the context of KD and MPS I. It provides an overview of tolerance regions of univariate and bivariate normal distributions, their estimation by prediction regions, and a roadmap for using them to develop BVNL NBS tests for additional inherited metabolic diseases.

The accuracy of BVNL tests to be developed for additional diseases, of course, will depend on identifying suitable biomarkers, in addition to the enzymes whose deficiency is the primary cause of disease, that reflects well the buildup of toxic metabolites in cells that contribute to the pathogenesis of the diseases. In the case of KD and MPS I, PSY and HS worked extremely well as secondary biomarkers. The potential for similarly informative secondary biomarkers to improve NBS tests for at least some other inherited metabolic diseases seems to warrant continued research.

One caveat with respect to MPS I is the observation that a more stringent univariate threshold for IDUA, corresponding to a value to the left of the left-most pseudo-deficient, but to the right of the MPS I cases would also discriminate the known cases from the normal observations without the need for a BVNL approach. However, given the small samples, it is possible that a true MPS I case may be missed using only a univariate approach if that case has sufficiently elevated HS.

Similarly with PD, a GAA threshold of −7.5 would eliminate all false positive cases. However, future PD cases may be missed. Given the high-risk for these newborns, a higher GAA threshold to allow more potential false positives coupled with genetic testing may be the optimal strategy. Continued development of the PD ellipse with other biomarkers or statistical transformations may improve efficiency, but in its current state would outperform the use of univariate thresholds for GAA and CRE. Additional variables such as birth weight, age, transfusion status, etc. may serve to improve accuracy as well.

There remain open questions as to how the BVNL methodology would be integrated into NBS program workflow efficiently, as well as measurement differences between laboratories affecting the consistency of results. Regarding the former, one advantage this method has over other post-analytical tools is that these methods are not proprietary and each NBS program can decide how best to integrate the BVNL within the program workflow. It is likely that different programs may have different implementation strategies, and should BVNL methods be widely adopted these implementations will need to be evaluated. As to the latter, while standardization of the biomarkers provides easily comparable values, laboratory differences may still exist [65]. Maintaining distinct BVNL tools for each laboratory is one potential solution, although a unified approach would be preferable.

We propose BVNL methodology as an effective way to improve NBS programs and expect that the advantage of the BVNL test over a bivariate test that predicts disease if X < τ_1_ and Y > τ_2_ seen in our KD studies will prove to be a general phenomenon. Due to the additional information found in the bivariate relationship between the biomarkers, BVNL tests can be expected to have lower false positive rates than bivariate alternatives without notably increased false negative rates. This property of BVNL tests suggest that existing screening tools that utilize univariate thresholds may see improvement by implementing bivariate or multivariate normal tests wherever univariate tests are used. Prospective testing of BVNL methodology would consequently appear to be warranted.

## Figures and Tables

**Figure 1 IJNS-08-00061-f001:**
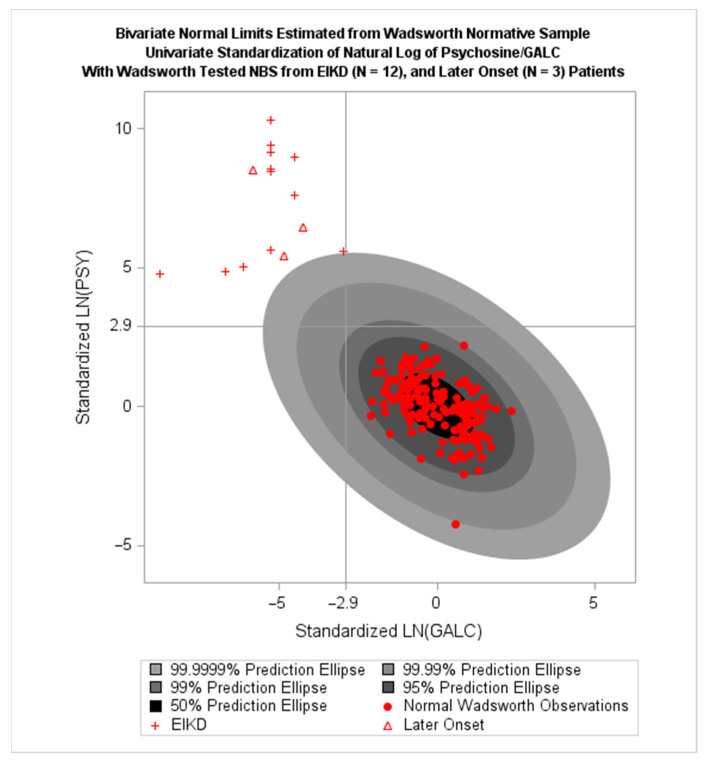
BVNL estimated from Wadsworth Normative Sample (red circles; N = 166), with Wadsworth tested NBS from EIKD (red crosses; N = 12) and Later Onset (red triangles; N = 3) Age at onset ranged from birth to six months in EIKD and 11 to 29 months in Later Onset. Points in the upper left quadrant and outside the ellipse represent positive screens for KD.

**Figure 2 IJNS-08-00061-f002:**
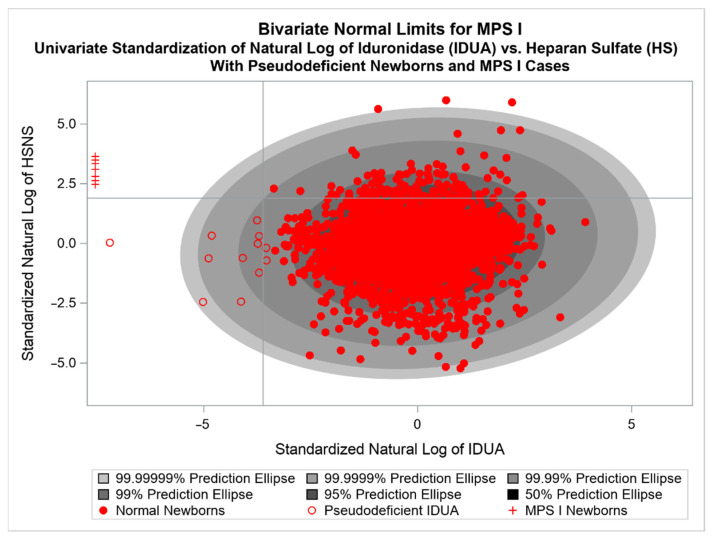
BVNL estimated from Gifu Normative Sample (open and filled red circles; N = 5000) based on alpha-L-iduronidase (IDUA) and Heparan Sulfate (HS) level. All MPS I newborns (red plus signs, N = 7) and 12 pseudo-deficient cases (open red circles, N = 12) are correctly identified.

## Data Availability

The data presented in this study are available on request from the corresponding author. The data are not publicly available due to privacy concerns.

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
