# Peer review of "A Roadmap for Potential Improvement of Newborn Screening for Inherited Metabolic Diseases Following Recent Developments and Successful Applications of Bivariate Normal Limits for Pre-Symptomatic Detection of MPS I, Pompe Disease, and Krabbe Disease"

_2409-515X, 2022, doi:10.3390/ijns8040061_

Round 1

Reviewer 1 Report

Dear authors,

Thank you for this opportunity of reviewing your paper. 

In my country, (Japan), it has not been started yet official NBS including MPS1, KD and PD. This paper help to establish the screening level about those diseases. I think this is suitable to accept.

There are some question and request to understand well for the non-specialists.

What about the difference between the measuring machines?

This result is very interesting results, but please mention the limitations of the study.

Author Response

Dear reviewer,

Thank you for your helpful response.  Regarding potential measurement differences, the BVNL approach standardizes measurements so that differences should be minimized.  However, it may be the case that each laboratory should have its own set of tools.  The updated manuscript has in its second to last paragraph a brief discussion of these points.

Thanks again for your comments.

Sincerely,

Kabir Jalal

Reviewer 2 Report

The manuscript by Kabir, et al illustrates the potential use of BVNL in NBS programs to reduce FPR and improve PPV.

The following comments and suggestions seek to improve the manuscript:

1) Replace the term 'mutation' with the more accepted nomenclature of 'pathogenic variant' throughout

2) Ensure gene names are italicized

3) The ACHDNC has never removed a condition, so would remove that indication. Likewise, the ACHDNC is a recommendation only, and not a requirement. In addition, please be clear to indicate that NBS programs do not screen FOR secondary conditions. They only screen for the core conditions and may or may not detect secondary conditions through this process.

4) There has been much work looking at potential anxiety stemming from FP NBS results with various findings - many showing that pervasive anxiety does not actually occur. Would look into these and soften the language and reference these around the impacts of FP results. (

5) Overall, I thought that the background was much too exhaustive for the purpose of this paper. I do not think that such robust description of the disease course and treatment is necessary for this. The important point is the genetic and enzymatic cause of the disease and the need for early detection in order to administer treatment earlier.

6) The section on NBS history and current practices seems to rely on quite old data. This needs to be updated to reflect the actual current practices. There are over 20 states now screening for MPS I and Pompe in the US. Likewise, many are now using 2nd and 3rd tier screening models which have much improved the FPR and PPV that were originally published years ago. Utilizing these references suggests that the current problem is bigger than it actually is - owing to adoption of 2nd tier GAG and PSY and ratio analyses as well as 3rd tier molecular sequencing. GAG analysis and accuracy may also vary by method used, and has been greatly enhanced in recent years.

7) It is mentioned that MPS I is the only MPS on the RUSP, but MPS II has recently been added.

8) It was not abundantly clear to me how BVNL would actually be implemented into a high-throughout NBS program workflow. This has been an issue with utilization of other post-analytical tools like CLIR and Random Forest/ML tools. Additionally, it is not clear as to the importance of integrating additional variables that often skew NBS results, including age at time of specimen collection, transfusion status, gestational age, birth weight, etc.

Author Response

Dear reviewer,

Thank you for taking the time to review our manuscript.  Your suggested changes were helpful and we have tried to address them in this revised paper as detailed below.

  1. Pathogenic variant has replaced mutation throughout the manuscript (pg 3, paragraph 3, 3 instances; pg 5, last paragraph).
  2. Gene names have been italicized (pg 3, paragraph 3).
  3. We have clarified that the ACHDNC does not remove conditions, and that each state screening program is free to ignore their recommendations.
  4. A paragraph discussing familial anxiety related to false positives has been added (pg 2, paragraph 5).
  5. In section 2.1, much of the background related to disease course and treatment has been removed to focus manuscript (section 2.1).
  6. The recent addition of MPS II to the RUSP has been acknowledged, with references to the decision letters (pg 5, last paragraph in section 3.1).
  7. The paragraphs detailing the performance history of newborn screening tools has been revised, and a greater emphasis has been placed on more recent results.

Thanks again for your comments.

Sincerely,

Kabir Jalal